# Correlation of Spectral CT-Based Iodine Concentration Parameters with LI-RADS Classification of Suspected Hepatocellular Carcinoma Nodules in Cirrhotic Patients

**DOI:** 10.3390/diagnostics15060725

**Published:** 2025-03-14

**Authors:** Antonio Celestino, Paolo Marra, Alessandro Barbaro, Carlotta Gargiulo, Riccardo Muglia, Giuseppe Muscogiuri, Pietro Andrea Bonaffini, Sandro Sironi

**Affiliations:** 1School of Medicine, University of Milano-Bicocca, 20126 Milan, Italy; a.celestino1@campus.unimib.it (A.C.); a.barbaro1@campus.unimib.it (A.B.); c.gargiulo4@campus.unimib.it (C.G.); rmuglia@asst-pg23.it (R.M.); gmuscogiuri@asst-pg23.it (G.M.); pbonaffini@asst-pg23.it (P.A.B.); sandro.sironi@unimib.it (S.S.); 2Department of Radiology, ASST Papa Giovanni XXIII Hospital, 24127 Bergamo, Italy

**Keywords:** hepatocellular carcinoma, iodine maps, liver cirrhosis, spectral computed tomography, diagnostic imaging

## Abstract

**Background**: The LI-RADS classification is widely used for the hepatocellular carcinoma (HCC) risk stratification of liver nodules in cirrhotic patients. The evaluation of nodule enhancement, which is a major criterion, commonly relies on qualitative assessment. This study aims to investigate the potential role of material density (MD) parameters in the iodine maps of spectral computed tomography (SCT) to discriminate between LI-RADS (v2018 CORE) categories in cirrhotic patients. **Methods**: Dual-energy SCT scans of cirrhotic patients with suspected HCC, taken between March 1st, 2022 and September 30th, 2023, were retrospectively reviewed. All the images were reviewed by trained radiologists to classify nodules as LI-RADS 3, 4, or 5 by consensus. MD maps were generated in the hepatic arterial phase (HAP), portal venous phase (PVP), and equilibrium phase (EP). The iodine concentration density (ICD) values of nodules (ICDnodule) and the non-nodular liver parenchyma (ICDliver) were measured to calculate lesion-to-non-nodular liver ICD ratio (LNR), as well as their differences (ΔICD) and ratios (rLNR). Results were correlated with LI-RADS categories. **Results**: A total of 69 patients were included and 79 DECT exams were assessed. Overall, 197 nodules (size 24.67 ± 23.11 mm, mean ± SD) were categorized into different LI-RADS classes: 44 were classed as LI-RADS 3 (22.3%), 14 were classed as LI-RADS 4 (7.1%), and 139 were classed as LI-RADS 5 (70.6%). The arterial LNR, arterial ICDnodule, ΔICD, and rLNR between HAP and PVP discriminated between LI-RADS 3 and LI-RADS 4+5 nodules (*p* < 0.001). All the calculated MD parameters showed high diagnostic accuracy rates (all AUCs = 70–73%). **Conclusions**: MD parameters of liver nodules measured in SCT scans are viable diagnostic tools that may increase the radiologist’s confidence in LI-RADS class allocation in cirrhotic patients. This preliminary and speculative study can serve as a baseline for the potential quantification of iodine concentrations of focal liver lesions to reduce subjectivity in hepatic nodule assessment and reporting. Future perspectives include the quantification of iodine concentration for prognostic stratification before locoregional and systemic treatments in HCC patients.

## 1. Introduction

Since its development in 2011, the Liver Imaging Reporting and Data System (LI-RADS, v2018) had aimed to standardize the assessment of CT and MR imaging for HCC [1]. Two recent meta-analyses [2,3] contributed to the validation of the LI-RADS system, demonstrating that higher LI-RADS categories correlate with increasing proportions of HCC and overall malignancy, as anticipated.

Spectral CT (SCT) is a novel imaging approach, with enhanced tissue characterization capabilities compared to conventional single-energy CT (sECT). By capturing images from two different x-ray spectra, respectively, at low (70–80 kVp) and high energy levels (120–140 kVp), SCT improves the visibility of lesions without increasing contrast volume by generating virtual monoenergetic imaging (VMI) reconstructions at low energy levels [4]. Additionally, SCT generates material density (MD) maps, enabling the differentiation and quantification of tissue components and materials with varying attenuation properties across different energy levels [5].

Based on these capabilities, several studies in the last decade reported that iodine concentration density potentially supports the diagnosis of different types of neoplasms [6,7,8] or may even predict treatment efficacy [9,10]. However, regarding the characterization of liver nodules in cirrhotic patients with suspected HCC, the scientific evidence supporting the quantitative analysis of spectral CT remains limited.

Currently, no study in the literature has investigated any possible quantitative relationship between the MD parameters derived from iodine maps and the LI-RADS classification. On these basis, the present study aims to investigate the possibility of defining MD parameters derived from SCT using the iodine maps to discriminate among LI-RADS categories of nodules in cirrhotic patients in order to increase the radiologist’s confidence in LI-RADS class allocation.

## 2. Materials and Methods

### 2.1. Study Population

This retrospective study was performed in a single referral centre for liver transplants in accordance with the Declaration of Helsinki. Approval was obtained from the ethics committee of Bergamo (ref. n. 132-21, June 21st 2021).

The primary endpoint of this study was defining those MD parameters derived from SCT, using the iodine maps that discriminate among LI-RADS categories of nodules in cirrhotic patients, with the aim of increasing the radiologist’s confidence in LI-RADS class allocation.

All the adult cirrhotic patients, with 1 or more suspected liver lesion detected in abdominal CT scans performed between March 1st 2022 and September 30th 2023 and with at least 6 months of follow-up available, were included in this study.

The study’s design and workflow are summarized in Figure 1. Only CT scans performed with dual-energy technique were included, with the detection of nodules reported as LI-RADS 3-5 [1]. In the case of multifocal disease, a maximum of 5 nodules per patient were included in the analysis, selected in descending order by size (from 4 mm).

The exclusion criteria were as follows: liver nodules showing non-hepatocellular lesions (LR-Ms); atypical hypovascular/hypodense nodules; and nodules relapsing at the site of locoregional treatments.

### 2.2. DECT Protocol

All selected cases were performed using a single-source dual-energy CT scanner, equipped with a 256-cell detector with a 0.625 mm pitch, along the patient’s longitudinal axis and with fast kVp switching between 80 and 140 kVp (Revolution CT, General Electric; Waukesha, WI, USA). We also performed a standard quadriphasic abdominal scan. In the study period, the equilibrium phase was acquired for all the patients, but with dual-energy mode in a small proportion of cases. From dual-energy acquisitions, iodine maps were then obtained for each scan in the arterial, portal venous and, when available, equilibrium phases.

Iomeprol 400 mg/mL (Iomeron, Bracco, Milan, Italy) was used as a contrast agent and was intravenously injected in an antecubital vein at the rate of 4 mL/s and at a dose of 1.5 mL/kg of body weight. Dose adjustments (20% drop off) were considered in cases of renal insufficiency, as per protocol. The arterial phase started with the bolus tracking technique when abdominal aorta density reached the threshold of 150 HU. The portal venous phase started with a delay of 35 s from the arterial phase, while the equilibrium venous phase was acquired 3 min after contrast injection.

The scanning protocol and technique are summarized in Table 1.

An advanced deep learning-based image reconstruction algorithm, named TrueFidelity, was used.

### 2.3. Image Analysis

Image analysis was independently performed by two 3rd-year radiology residents. First, qualitative analysis was performed: by reviewing the original radiological report, every nodule was assigned to an LI-RADS category, validated by a board-certified radiologist. If no consensus was obtained for the original report, images were reviewed by a third independent radiologist experienced in hepatobiliopancreatic imaging. All the nodules were classified as LR-3 (intermediate probability), LR-4 (probably HCC), or LR-5 (definitely HCC) [1]. For the purpose of the statistical analysis, all the nodules were divided into two main groups: probably–definitely HCC (LR-4 and LR-5) versus an intermediate probability of HCC (LR-3).

The following parameters were extrapolated. We measured the iodine concentration density of the target nodule(s) (ICDnodule), the non-nodular hepatic liver parenchyma (ICDliver) and aorta (ICDaorta) on the material density maps (iodine–water maps) in the hepatic arterial phase (HAP), the portal venous phase (PVP) and, when available, the equilibrium phase (EP). All ICDs are expressed as mg/mL. Measurements of ICDnodule were obtained by manually placing a rounded 5 mm region of interest (ROI) in the most avidly and homogeneously enhancing region of the nodule(s), excluding areas of relative hypodensity and heterogeneity, in the HAP (aICDnodule), in the PVP (vICDnodule), and in the EP (eICDnodule) (Figure 2). In co-registered venous phases, the ROI was placed in exact correspondence with the one positioned in the HAP. Another ROI was manually placed in the lumen of the aorta, by convention on the same plane of the origin of the celiac trunk, respectively, in the HAP (aICDaorta), in the PVP (vICDaorta), and in the EP (eICDaorta). Lastly, three different circular ROIs were placed in the non-nodular hepatic parenchyma, excluding areas of heterogeneity or vessels, and then the average value was calculated in the HAP (aICDliver), in the PVP (vICDliver) and in the EP (eICDliver).

Using the aforementioned values, the following parameters were derived:

(1) LNR or lesion-to-non-nodular liver ratio in the HAP (aLNR = aICDnodule/aICDliver), in the PVP (vLNR = vICDnodule/vICDliver), and in the EP (eLNR = eICDnodule/eICDliver);

(2) NICD or normalized iodine concentration with aorta in the HAP (aNICD = aICDnodule/aICDaorta), in the PVP (vNICD = vICDnodule/vICDaorta), and in the EP (eNICD = eICDnodule/eICDaorta);

(3) ΔICD or difference in ICDs from the HAP to PVP (ΔavICD = aICDnodule − vICDnodule) and from the HAP to EP (ΔaeICD = aICDnodule − eICDnodule);

(4) Ratio between LNR in the HAP and in the PVP (ravLNR = aLNR/vLNR) and in the HAP and in the EP (raeLNR = aLNR/eLNR).

The lesion-to-non-nodular liver ratio was calculated to quantify the enhancement behaviour of the nodules, while the normalization with the aorta was performed as a control. All the derived parameters are summarized in Table 2.

### 2.4. Statistical Analysis

All the continuous variables were reported as mean ± standard deviation (SD) for data with a normal distribution and as the median with interquartile range (IQR) for data without a normal distribution.

Mann–Whitney U tests were used to compare LI-RADS classes (LI-RADS 3 vs. LI-RADS 4+5) for all MD parameters obtained from arterial, portal venous, and equilibrium phases on SCT scans.

Receiver operating characteristic (ROC) analysis was performed to assess the capacity of MD parameters to discriminate between LI-RADS classes; ROC curves with a corresponding area under the curve (AUC) were shown and the best cutoffs were calculated using the Youden index. Sensitivity and specificity were reported with their 95% confidence intervals (CIs). Finally, logistic regressions were performed to test the goodness of cutoffs to discriminate LI-RADS class; odds ratio with 95% CI were reported.

A *p*-value of less than 0.05 was considered statistically significant. All the analyses were performed with R software (version 4.3.0).

## 3. Results

### 3.1. Study Population

Among the 380 patients examined, 69 patients (M:F = 53:16) were eligible and included in the study. The median age of the patients was 68 years (IQR 61–76 years).

### 3.2. HCC Nodules Features

A total of 197 nodules in 79 DECT exams were examined (2.9 nodules per patient, range 1–5). Overall, 31 of 79 DECT exams acquired the equilibrium phase in dual-energy mode, for a total of 74/197 nodules (37.56%). They were categorized in different LI-RADS classes, as follows: 44 nodules were classified as LI-RADS 3 (22.34%), 14 were classified as LI-RADS 4 (7.11%), and 139 were classified as LI-RADS 5 (70.56%).

The mean size of the nodules was, respectively, 24.26 ± 22.07 mm (mean ± SD; range 4–137 mm) as regards all the nodules (*n* = 197), 11.57 ± 3.71 mm (mean ± SD; range 4–19 mm) as regards LI-RADS 3 nodules (*n* = 44), and 29.32 ± 23.77 mm (mean ± SD; range 6–137 mm) as regards LI-RADS 4+5 nodules (*n* = 153).

### 3.3. MD Parameters

As shown in Table 3, most of the quantitative parameters showed statistically significant values (*p* < 0.001) among LIRADS 4-5 and LIRADS 3 lesions: the only parameters that showed similar values between the two groups were, respectively, vLRN, vNICD, vICDnodule, dLRN, eNICD, and eICDnodule. Furthermore, when considering only those parameters that simultaneously include data from arterial to portal venous phases (namely ΔavICD and ravLNR), stronger statistical significance (*p* < 0.0001) was observed.

The median value of aICDnodule was significantly higher (*p* = 0.0002) for LI-RADS 4-5 nodules (31 mg/mL; IQR 27–39 mg/mL) compared to LI-RADS 3 (25.5 mg/mL; IQR 19–32 mg/mL) nodules. Moreover, the aLNR was significantly greater (*p* = 0.0001) for LI-RADS 4-5 nodules (2.3; IQR 1.8–3.2) compared to LI-RADS 3 (1.61; IQR 1.1–2.3) nodules.

Concerning the relationship between the arterial and the portal venous phases, the ΔavICD was significantly higher (*p* < 0.0001) for LI-RADS 4-5 nodules (8 mg/mL, IQR 3–14 mg/mL) compared to LI-RADS 3 (0 mg/mL, IQR −6–7 mg/mL) lesions. The ravLNR values were slightly higher for LI-RADS 4-5 nodules (median value 2.6; IQR 2–3.4) compared to LI-RADS 3 (median value 1.8, IQR 1.3–2.3) lesions (*p* < 0.0001).

The MD parameters from the arterial to the equilibrium phase had a similar behaviour: in particular, the ΔaeICD was moderately higher for LI-RADS 4-5 nodules (median value 13 mg/mL; IQR 6–20 mg/mL) compared to LI-RADS 3 (median value 6 mg/mL; IQR 3–10.8 mg/mL) ones (*p* = 0.0076), and raeLNR was slightly higher for LI-RADS 4-5 nodules (median value 2.6; IQR 2–3.4) versus LI-RADS 3 (median 2.4; IQR 1.8–3.2) nodules (*p* = 0.0001).

### 3.4. ROC and Cutoff Definition

All the statistically significant values are graphically represented in Figure 3 and corresponding receiver operating curves (ROC) of MD parameters with the area under the curve (AUC) are shown in Figure 4.

Table 4 shows the cutoff values for various MD parameters identified through ROC curves along with the corresponding AUC, while Table 5 shows the nominal logistic regression of the same MD parameters. Nominal logistic regression (Table 5) showed that aLNR > 1.39 with an odds ratio (OR) of 9.81 (95% CI 4.25–23.79) correctly classified LI-RADS 4-5 nodules in 85% cases (*p* < 0.0001), ΔavICD > 1 with an odds ratio (OR) of 7.07 (95% CI 3.41–15.04) correctly classified LI-RADS 4-5 nodules in 87.2% cases (*p* < 0.0001), and ravLNR > 1.87 with an odds ratio (OR) of 6.62 (95% CI 3.24–13.97) correctly classified LI-RADS 4-5 nodules in 88.3% cases (*p* < 0.0001). Similarly, aICDnodule > 28 mg/mL with an odds ratio (OR) of 4.06 (95% CI 2.03–8.33) was consistent for LI-RADS 4-5 nodules in 86.6% cases (*p* = 0.0001).

## 4. Discussion

This study analyzed several parameters related to iodine concentration density (ICD), respectively, in the hepatic focal lesion (ICDnodule), in the non-nodular liver parenchyma (ICDliver), and in the aorta (ICDaorta), and measured these in the available CT dynamic phases: the hepatic arterial phase (HAP), the portal venous phase (PVP), and the equilibrium phase (EP). Derived parameters were calculated for each phase: the lesion-to-non-nodular liver ratio (LNR) was determined to quantify the enhancement behaviour of the nodules and the normalized iodine concentration with the aorta (NICD) as a control. Additionally, we assessed differences in ICDs between the arterial and the portal venous phase (ΔavICD) and between the arterial and the equilibrium phase (ΔaeICD) and examined the LNR between the arterial and the portal venous phase (ravLNR) as well as the LNR between the arterial and the equilibrium phase (raeLNR): these parameters further provided valuable insights into the dynamic changes in iodine concentration in liver lesions, assisting in their possible characterization.

The most relevant finding of this preliminary study is that quantitative MD parameters derived from SCT studies (particularly aLNR, aICDnodule, ΔavICD and ravLNR) may discriminate between different LI-RADS categories of liver nodules in cirrhotic patients. Some of these parameters, namely ΔavICD and ravLNR, had stronger statistical significance. Moreover, different LI-RADS classes assigned by expert radiologists indeed exhibit statistically different parameters derived from SCT. All the above MD parameters have an overall similar, or even slightly higher, diagnostic accuracy (overall AUCs = 70–73%) compared to those previously reported by non-spectral CT (up to 70.5%) [11].

These results are closely in line with similar data from the literature [12]. Namely, Gao et al. [13] demonstrated that aLNR was the best MD parameter used in SCT scans for differentiating HCC from other liver nodules in 46 patients for a total of 23 nodules, while Pfeiffer et al. [14] reported similar efficiency using aLNR, by making a comparison with the MRI characteristics of the nodules. It was even a better predictor than standard MRI when characterizing HCC. Some other studies focused on assessing different parameters for differentiation between hepatic hemangiomas and HCC [15], or assessed the correlation between iodine concentrations and other features (i.e., iodized oil retention with tumour responses in HCC patients [16], perfusion CT [17], or histopathological findings [12]).

However, from the analysis conducted in this study, some MD parameters derived from the PVP were found to be less accurate then those calculate in the HAP. Particularly, vLNR did not show significant discrimination between nodules of different LI-RADS categories. Even if this observation requires further validation, the use of MD parameters on the HAP seems more promising compared to their used on PVP. Similarly, MD parameters in the equilibrium phase have demonstrated an even higher diagnostic accuracy compared to PVP parameters (AUC 72% and 82%, respectively), although the small number of liver nodules evaluated in EP scans were acquired in dual-energy mode (74/197 nodules).

The analysis in this study highlighted that aLNR is one of the best objective predictors for discriminating between the LI-RADS 3 and LI-RADS 4-5 categories of liver nodules. Indeed, both ΔavICD and ravLNR appeared to be the most reliable markers, with the highest diagnostic accuracies (AUCs of 72% and 73%, respectively). The most plausible explanation for this is their incorporation of data from both HAP and PVP, thus integrating the vascular behaviour of HCC nodules in the two different phases and allowing for more reliable distinct cutoff values for LI-RADS 3 nodules compared to LI-RADS 4-5 nodules.

This confirms the potential role of SCT in the diagnosis and characterization of liver nodules in cirrhotic patients, compared to the conventional Single-Energy CT [11].

Furthermore, the use of MD parameters obtained from DECT scans may also increase radiologists’ confidence in LI-RADS class allocation by simultaneously reducing CT scanner limitations, the use of contrast injection protocol, operator dependency, and hemodynamic changes. This could open new ways to approaching the follow-up and subsequently the treatment of HCC nodules: for example, according to these results, all liver nodules with aLNR values < 1.39 could be monitored for stricter surveillance as they may have a higher index of suspicion, while all nodules with classical imaging appearance of HCC or aLNR values > 1.39 can be considered to be immediately reported in multidisciplinary team’s discussion as all these nodules belong to the LI-RADS 4 or 5 category and have a higher risk of malignancy. As concerns clinical practice, those numerical cutoff values derived from SCT scans, in the current era of machine-based learning and of very fast progress in artificial intelligence, would both simplify and accelerate the algorithm for diagnosis of nodules in cirrhotic patients, with fewer ‘operator-dependent’ errors in the future. Plus, different MD values may also be helpful for obtaining a better characterization of other benign (e.g., fibro-nodular hyperplasia, haemangioma) and non-HCC malignant tumours in the liver.

This study has some limitations. First, the lack of an adequate number of LI-RADS 4 nodules did not allow us to consider it as a separate category, and this may have led to bias in the results. Other limitations are the lack of histopathological correlation (even if HCC can be diagnosed through imaging when overt according to EASL guidelines [18,19]); the exclusion of nodules treated with interventional radiology procedures and/or surgical resection (in order to reduce any bias derived from post-treatment changes); the exclusion of hypovascular/hypodense nodules; and the lack of several data points from follow-up SCT scans of all the patients due to liver transplant as well as death or loss during the follow-up. Furthermore, many factors may influence the biodistribution of the contrast agent and therefore the iodine concentration density. These factors are both intrinsic (such as variability in nodule size, nodule heterogeneity, and perfusion abnormalities due to a different grade of hepatic arterialization), and extrinsic (such as the patient’s cardiac function, vein quality affecting flows, or contrast quantity reduction in patients with renal insufficiency). For the statistical analysis in this study, all the nodules were divided into two main groups: probably–definitely HCC (LR-4 and LR-5) versus an intermediate probability of HCC (LR-3). This aligns with a recent meta-analysis which demonstrated that grouping LI-RADS 4 and 5 into a single category, as opposed to considering LI-RADS 5 alone, results in an approximately 12% increase in sensitivity [20]. Furthermore, this reflects the clinical practice of other centres where LR-4 and LR-5 lesions are managed operatively to achieve treatment at the earliest stage [21]. This approach aligns with the latest LI-RADS recommendations that suggest the multidisciplinary team’s decision for treatment of LR-4 nodules in selected patients [22].

As this study aimed to investigate the potential role of quantitative assessment in HCC characterization, using MD parameters, in order to increase the radiologists’ confidence in LI-RADS class allocation and the characterization of liver nodules in cirrhotic patients, these preliminary results highlighted three main MD parameters (aLNR, ΔavICD and ravLNR) and suggested possible cutoff values for the differentiation of different LI-RADS categories: for example, the eventual quantification of HCC nodules may allow for a very early diagnosis of those LI-RADS 3 nodules with a higher risk of shifting to an LI-RADS 4 or 5 category, with potential value in prognostication and treatment response assessment.

In conclusion, this is a preliminary and speculative study whose premises may serve as a baseline for a potential prospective quantification of the iodine concentration of focal liver lesions. Also, in a future perspective, long-term studies are needed to evaluate MD parameters from SCT scans of liver nodules for a better correlation between the identified cutoff values, the LI-RADS category, and prognosis.

## Figures and Tables

**Figure 1 diagnostics-15-00725-f001:**
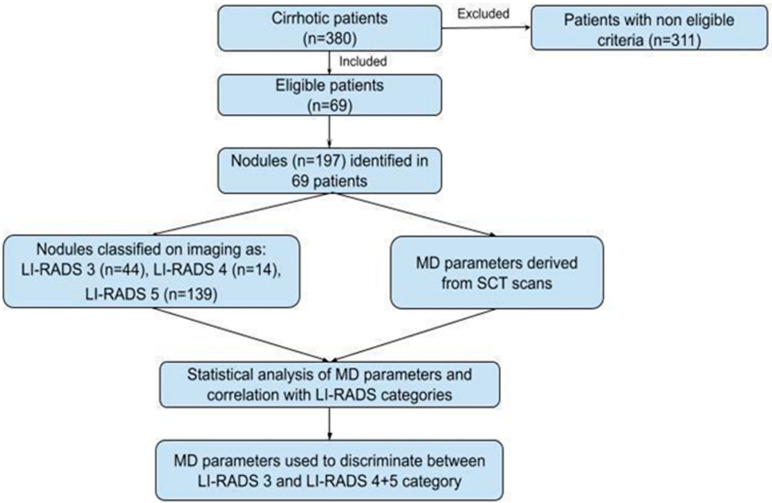
A flowchart of the study’s design and workflow.

**Figure 2 diagnostics-15-00725-f002:**
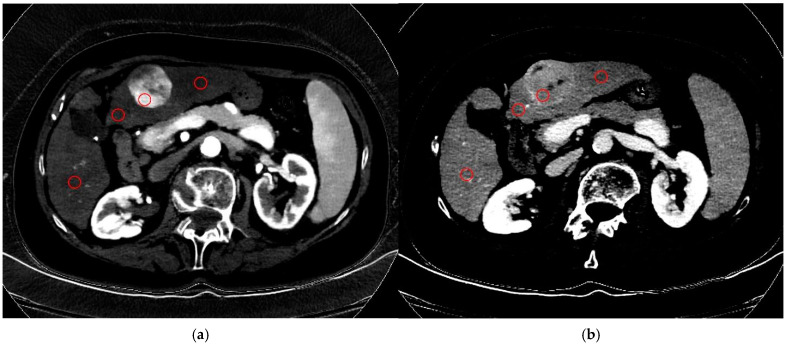
The spectral computed tomography (SCT) result of a patient with cirrhosis and hepatocellular carcinoma. (**a**). An iodine density map derived from the hepatic arterial phase (HAP) showing ROI within the lesion used to calculate the aICD_nodule_, three ROIs in the liver used to calculate aICD_liver_, and another ROI used to calculate aICD_aorta_ (**b**). An iodine density map derived from the portal venous phase (PVP) showing ROI within the lesion used to calculate vICD_nodule_, three ROIs in the liver used to calculate vICD_liver_, and another ROI used to calculate vICD_aorta_. ***HAP*** hepatic arterial phase, ***PVP*** portal venous phase, ***EP*** equilibrium phase, ***aICDaorta*** iodine concentration density of the aorta in the HAP, a***ICDliver*** iodine concentration density of non-nodular hepatic liver parenchyma in the HAP, a***ICDnodule*** iodine concentration density of the target nodule(s) in the HAP, ***vICDaorta*** iodine concentration density of the aorta in the PVP, v***ICDliver*** iodine concentration density of non-nodular hepatic liver parenchyma in the PVP, v***ICDnodule*** iodine Concentration Density of the Target Nodule(s) in the PVP.

**Figure 3 diagnostics-15-00725-f003:**
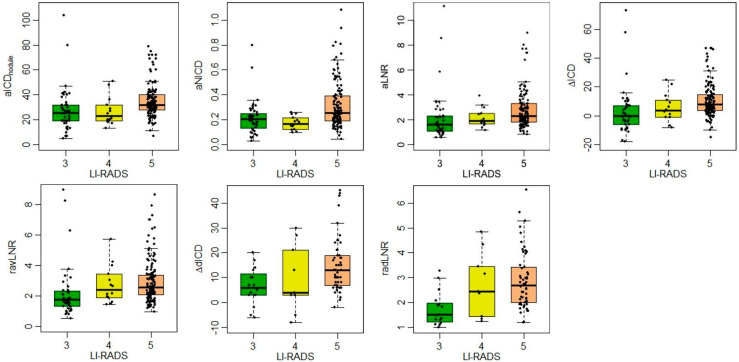
A boxplot of MD parameters with statistical significance according to LI-RADS classes (3-4-5). ***aICDnodule*** the iodine concentration density of the target nodule(s) in the hepatic arterial phase, ***aNICD*** the normalized iodine concentration with aorta in the hepatic arterial phase, ***aLNR*** the lesion-to-non-nodular liver ratio in the hepatic arterial phase, **Δ*ICD*** the difference in the iodine concentration density from the hepatic arterial phase to the portal venous phase, ***ravLNR*** the ratio between the lesion-to-non-nodular liver ratio in the hepatic arterial phase and the portal venous phase, **Δ*dICD*** the difference in iodine concentration density from the hepatic arterial phase to the equilibrium phase, ***radLNR*** the ratio between the lesion-to-non-nodular liver ratio in the hepatic arterial phase and equilibrium phase.

**Figure 4 diagnostics-15-00725-f004:**
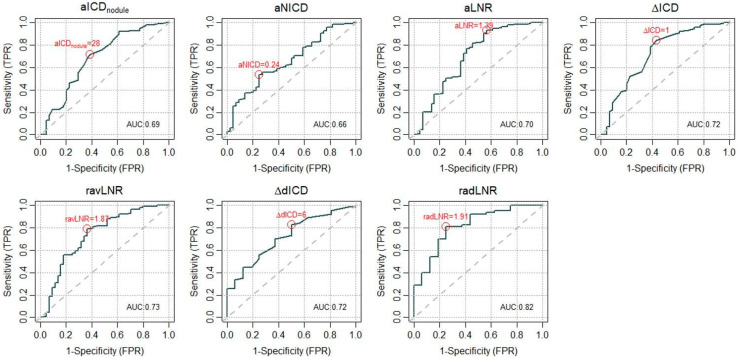
ROC curves and the AUC for MD parameters (statistically significant in Table 3) according to LI-RADS class (3 vs. 4-5). ***aICDnodule*** the iodine concentration density of the target nodule(s) in the hepatic arterial phase, ***aNICD*** the normalized iodine concentration with the aorta in the hepatic arterial phase, ***aLNR*** the lesion-to-non-nodular liver ratio in the hepatic arterial phase, **Δ*ICD*** the difference in iodine concentration density from the hepatic arterial phase to the portal venous phase, ***ravLNR*** the ratio between the lesion-to-non-nodular liver ratio in the hepatic arterial phase and the portal venous phase, **Δ*dICD*** the difference in iodine concentration density from the hepatic arterial phase to the equilibrium phase, ***radLNR*** the ratio between the lesion-to-non-nodular liver ratio in the hepatic arterial phase and equilibrium phase.

**Table 1 diagnostics-15-00725-t001:** DECT protocol.

Position	Supine
Direction	Craniocaudal
Contrast agent dose and rate	1.5 mL/kg body weight, at the rate of 4 mL/s
Saline bolus and rate	30 mL at the rate of 4 mL/s
Scan delay	Automated triggering (threshold at 150 HU in the aorta)
Slice thickness	0.625 mm
Reconstruction parameters	1 mm thickness

**Table 2 diagnostics-15-00725-t002:** List of parameters derived from iodine maps in arterial (a), portal venous (p) and equilibrium (e) phases.

aLNR = aICD_nodule_/aICD_liver_	lesion-to-non-nodular liver ratio in HAP
vLNR = vICD_nodule_/vICD_liver_	lesion-to-non-nodular liver ratio in PVP
eLNR = ICD_nodule_/eICD_liver_	lesion-to-non-nodular liver ratio in EP
aNICD = aICD_nodule_/aICD_aorta_	normalized iodine concentration with aorta in HAP
vNICD = vICD_nodule_/vICD_aorta_	normalized iodine concentration with aorta in PVP
eNICD = eICD_nodule_/eICD_aorta_	normalized iodine concentration with aorta in EP
ΔavICD = aICD_nodule_ − vICD_nodule_	difference in ICD from HAP to PVP
ΔaeICD = aICD_nodule_ − eICD_nodule_	difference in ICD from HAP to EP
ravLNR = aLNR/vLNR	ratio between LNR in HAP and in PVP
raeLNR = aLNR/eLNR	ratio between LNR in HAP and in EP

***ICDaorta*** iodine concentration density of the aorta, ***ICDliver*** iodine concentration density of non-nodular hepatic liver parenchyma, ***ICDnodule*** iodine concentration density of the target nodule(s), ***HAP*** hepatic arterial phase, ***PVP*** portal venous phase, ***EP*** equilibrium phase, ***NICD*** normalized iodine concentration with aorta, ***LNR*** lesion-to-non-nodular liver ratio.

**Table 3 diagnostics-15-00725-t003:** Median and IQR of MD parameters according to LI-RADS classes (3 vs. 4-5).

	All	LI-RADS 3 (*n* = 44)	LI-RADS 4-5 (*n* = 153)	*p*
aICD_nodule_	30 (25–38)	25.5 (19–32)	31 (27–39)	**0.0002**
aNICD	0.2 (0.2–0.3)	0.2 (0.1–0.2)	0.2 (0.2–0.4)	**0.0015**
aLNR	2.2 (1.6–3)	1.6 (1.1–2.3)	2.3 (1.8–3.2)	**0.0001**
vICD_nodule_	25 (21–28)	24.5 (20–29)	25 (21–27)	0.5179
vNICD	0.5 (0.5–0.6)	0.6 (0.4–0.6)	0.5 (0.5–0.6)	0.8571
vLNR	0.9 (0.8–1)	1 (0.8–1.1)	0.9 (0.8–1)	0.1726
ΔavICD = aICD_nodule_ − vICD_nodule_	7 (1–12)	0 (−6–7)	8 (3–14)	**<0.0001**
ravLNR = aLNR/vLNR	2.3 (1.8–3.2)	1.8 (1.3–2.3)	2.6 (2–3.4)	**<0.0001**
eICD_nodule_	19 (17–21.5)	17.5 (15–21)	19 (17–22)	0.1504
eNICD	0.6 (0.5–0.6)	0.5 (0.5–0.6)	0.6 (0.5–0.6)	0.2201
eLNR	0.9 (0.8–1)	0.9 (0.8–1)	1 (0.8–1)	0.1226
ΔaeICD = aICD_nodule_ − eICD_nodule_	10 (6–18)	6 (3–10.8)	13 (6–20)	**0.0076**
raeLNR = aLNR/eLNR	2.4 (1.8–3.2)	1.5 (1.3–1.9)	2.6 (2–3.4)	**0.0001**

***aICDnodule*** iodine concentration density of the target nodule(s) in the hepatic arterial phase, ***aNICD*** the normalized iodine concentration with the aorta in the hepatic arterial phase, ***aLNR*** the lesion-to-non-nodular liver ratio in the hepatic arterial phase, ***vICDnodule*** the iodine concentration density of the target nodule(s) in the portal venous phase, ***vNICD*** the normalized iodine concentration with aorta in the portal venous phase, ***vLNR*** the lesion-to-non-nodular liver ratio in the portal venous phase, **Δ*avICD*** the difference in iodine concentration density from the hepatic arterial phase to the portal venous phase, ***ravLNR*** the ratio between the lesion-to-non-nodular liver ratio in the hepatic arterial phase and the portal venous phase, ***eICDnodule*** the iodine concentration density of the target nodule(s) in the equilibrium phase, ***eNICD*** the normalized iodine concentration with the aorta in the equilibrium phase, ***eLNR*** the lesion-to-non-nodular liver ratio in the equilibrium phase, **Δ*aeICD*** the difference in the iodine concentration density from the hepatic arterial phase to the equilibrium phase, ***raeLNR*** the ratio between the lesion-to-non-nodular liver ratio in the hepatic arterial phase and the equilibrium phase.

**Table 4 diagnostics-15-00725-t004:** Sensitivity and specificity for all cutoff values.

	Sensitivity (95% CI)	Specificity (95% CI)
aICD_nodule_ = 28	0.72 (0.64, 0.79)	0.61 (0.45, 0.76)
aNICD = 0.24	0.52 (0.44, 0.60)	0.75 (0.60, 0.87)
aLNR = 1.39	0.93 (0.88, 0.96)	0.43 (0.28, 0.59)
ΔavICD = 1	0.84 (0.78, 0.90)	0.57 (0.41, 0.72)
ravLNR = 1.87	0.79 (0.72, 0.85)	0.64 (0.48, 0.78)
ΔaeICD = 6	0.83 (0.71, 0.91)	0.50 (0.25, 0.75)
raeLNR = 1.91	0.81 (0.69, 0.90)	0.75 (0.48, 0.93)

***aICDnodule*** the iodine concentration density of the target nodule(s) in the hepatic arterial phase, ***aNICD*** the normalized iodine concentration with the aorta in the hepatic arterial phase, ***aLNR*** the lesion-to-non-nodular liver ratio in the hepatic arterial phase, **Δ*avICD*** the difference in iodine concentration density from the hepatic arterial phase to the portal venous phase, ***ravLNR*** the ratio between the lesion-to-non-nodular liver ratio in the hepatic arterial phase and portal venous phase, **Δ*aedICD*** the difference in iodine concentration density from the hepatic arterial phase to the equilibrium phase, ***raeLNR*** the ratio between the lesion-to-non-nodular liver ratio in the hepatic arterial phase and equilibrium phase.

**Table 5 diagnostics-15-00725-t005:** Logistic regression for LI-RADS 4-5 vs. 3 according to categorized MD parameters.

	N LI-RADS 4-5 (%)	OR (95% CI)	*p*
**aICD_nodule_**			
<28	43 (61.4)	1	
>28	110 (86.6)	4.06 (2.03–8.33)	0.0001
**aNICD**			
<0.24	73 (68.9)	1	
>0.24	80 (87.9)	3.29 (1.59–7.25)	0.0019
**aLNR**			
<1.39	11 (36.7)	1	
>1.39	142 (85)	9.81 (4.25–23.79)	<0.0001
**ΔavICD**			
<1	24 (49)	1	
>1	129 (87.2)	7.07 (3.41–15.04)	<0.0001
**ravLNR**			
<1.87	32 (53.3)	1	
>1.87	121 (88.3)	6.62 (3.24–13.97)	<0.0001
**ΔaeICD**			
<6	11 (57.9)	1	
>6	52 (86.7)	4.73 (1.46–15.74)	0.0096
**raeLNR**			
<1.91	12 (50)	1	
>1.91	51 (92.7)	12.75 (3.75–52.41)	0.0001

***aICDnodule*** the iodine concentration density of the target nodule(s) in the hepatic arterial phase, ***aNICD*** the normalized iodine concentration with aorta in the hepatic arterial phase, ***aLNR*** the lesion-to-non-nodular liver ratio in the hepatic arterial phase, **Δ*avICD*** the difference in iodine concentration density from the hepatic arterial phase to the portal venous phase, ***ravLNR*** the ratio between the lesion-to-non-nodular liver ratio in the hepatic arterial phase and portal venous phase, **Δ*aedICD*** the difference in iodine concentration density from the hepatic arterial phase to the equilibrium phase, ***raeLNR*** the ratio between the lesion-to-non-nodular liver ratio in the hepatic arterial phase and equilibrium phase.

## Data Availability

No new data were created or analyzed in this study. Data sharing is not applicable to this article.

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
