# Peer review of "Correlation of Spectral CT-Based Iodine Concentration Parameters with LI-RADS Classification of Suspected Hepatocellular Carcinoma Nodules in Cirrhotic Patients"

_diagnostics, 2025, doi:10.3390/diagnostics15060725_

Round 1
Reviewer 1 Report
Comments and Suggestions for Authors
The author’s choice of topic is highly relevant. Since the global burden of hepatocellular carcinoma as a consequence of liver cirrhosis is large, primarily owing to the increasing rate of metabolic associated steatotic liver disease and alcohol-related liver disease, beside chronic hepatitis B and C virus infection. Supplemented to the widely used LIRADS classification system for HCC risk stratification, this study aimed to investigate the potential role of material density parameters in the iodine maps of Spectral Computed Tomography (SCT) to discriminate between LIRADS categories in cirrhotic patients.
Also, the application of artificial intelligence, the fast progress of machine-based learning the detection and classification of focal liver lesions using SCT is an exciting field. These technologies offer promising opportunities to enhance diagnostic accuracy and reduce variability in radiological interpretations.
General comments:
The manuscript is clear, relevant for the field of radiological assessment of HCC in cirrhotic liver and draw the attention for the potential role of SCT in the diagnosis and characterization of liver nodules, compared to the conventional single energy CT.
The methodology has been delineated with accuracy about patient selection, processing and identification of radiological features.
Statistical analysis seemed to be sophisticated, and adheres to the ethical principles and standards of the Declaration of Helsinki.
The description of the "Results" is clear, describing with well-illustrating figures and tables with the explanation of listed of derived parameters, median and IQR ranges of MD parameters according LIRADS classes. The figures of Boxplots and ROC curves also enhance understanding the results.
The “Discussion” places the findings in a broader context and provides several relevant international findings related to the work.
Suggestion:
The current (February 2025) HCC guideline might be cited in the article:
European Association for the Study of the Liver. EASL Clinical Practice Guidelines on the management of hepatocellular carcinoma. J Hepatol. 2025 Feb;82(2):315-374.
doi: 10.1016/j.jhep.2024.08.028.
Questions:
Among the measured quantitative MD parameters how can you explain the stronger statistical significance in case of considering only those parameters that simultaneously include data from arterial to portal venous phases (ΔavICD and ravLNR)?
Could the MD parameters obtained from DECT scans also be used in case of focal lesions suspect for malignancy without cirrhosis? It would be helpful for the diagnostic of HCC in MAFLD.
Overall Recommendation:
Accept after Minor Revisions: The paper can in principle be accepted after revision based on the reviewer’s comments.
Author Response
Comment #1: The current (February 2025) HCC guideline might be cited in the article: European Association for the Study of the Liver. EASL Clinical Practice Guidelines on the management of hepatocellular carcinoma. J Hepatol. 2025 Feb;82(2):315-374. doi: 10.1016/j.jhep.2024.08.028.
Response #1: Thank you for your appropriate suggestion: we did insert the proposed citation and renumbered the reference list accordingly.
Comment #2 Among the measured quantitative MD parameters how can you explain the stronger statistical significance in case of considering only those parameters that simultaneously include data from arterial to portal venous phases (ΔavICD and ravLNR)?
Response#2: The stronger statistical significance observed when considering parameters that include data from both the arterial and portal venous phases (ΔavICD and ravLNR) can be explained by the fact that these quantitative parameters reflect the dynamic behavior of the nodules in the two phases (namely, arterial wash-in and portal venous wash-out). This lastly reflects the standard behavior behind LIRADS classification system. Therefore, this might improve the ability to differentiate between LI-RADS 3 and LI-RADS 4+5 nodules. Specifically, ΔavICD provides a quantitative measurement of iodine concentration density changes between the arterial phase and the portal venous phase, while ravLNR enhances these changes as compared to non-nodular liver parenchyma.
Comment #3: Could the MD parameters obtained from DECT scans also be used in case of focal lesions suspect for malignancy without cirrhosis? It would be helpful for the diagnostic of HCC in MAFLD.
Responde#3: It is known that patients with MAFLD are at risk of developing HCC, and the risk is closely linked to the stage of fibrosis. Theoretically, we might expect that the same results obtained in cirrhotic patients could be seen in HCC arising in MAFLD patients, being the contrast behaviour superimposable. Further research is needed to clarify the role of Spectral CT in these patients and determine whether it could also enhance early detection strategies.
Reviewer 2 Report
Comments and Suggestions for Authors
This is a very interesting paper, but it contains some problems (see below), and is not acceptable in the present form.
1. Please add a list of abbreviations.
2. Parameters: Please add a brief comment of each parameter in Discussion.
3. Analysis of the Results: Please analyze the results from (supposed) hemodynamical or histological viewpoints.
Author Response
Comment #1: Please add a list of abbreviations.
Response #1: We updated the paper with all abbreviations. Moreover, we updated the figures and tables annotations, reporting all cited abbreviations used in the main manuscript.
Comment #2: Parameters: Please add a brief comment of each parameter in Discussion.
Response #2: We updated the manuscript in the Discussion session, accordingly: “This study analysed several parameters related to iodine concentration density (ICD), respectively in the hepatic focal lesion (ICDnodule), in the non-nodular liver parenchyma (ICDliver), and in the aorta (ICDaorta), and measured in the available CT dynamic phases: hepatic arterial phase (HAP), portal venous phase (PVP), and equilibrium phase (EP). Derived parameters were calculated for each phase: the lesion-to-non-nodular liver ratio (LNR) to quantify the enhancement behaviour of the nodules and the normalised iodine concentration with the aorta (NICD) as control. Additionally, differences in ICD between the arterial and the portal venous phase (ΔavICD) and between the arterial and the equilibrium phase (ΔaeICD) and LNR ratio between the arterial and the portal venous phase (ravLNR) and LNR ratio between the arterial and the equilibrium phase (raeLNR) were assessed: these parameters further provided valuable insights into the dynamic changes of iodine concentration in liver lesions for their possible characterization”.
Comment #3: Analysis of the Results: Please analyze the results from (supposed) hemodynamical or histological viewpoints.
Response #3: As demonstrated by numerous studies, the hepatocarcinogenetic microenvironment is a dynamic system, characterised by multiple molecular and histopathological changes that occur over time. Tajima et al. (https://doi.org/10.2214/ajr.178.4.1780885), in their study on sequential hemodynamic changes in HCC and dysplastic nodules, highlighted that, during the evolution of the hepatic nodule, the well-known arterial hyperperfusion observed in HCC nodules is preceded by a progressive depletion of native arterial vessels. This reduction leads to a transient rarefaction of the arterial supply, which could explain a temporary reduction in iodine concentration in the nodule. Additionally, portal flow depletion seems to occur more rapidly than arterial neoangiogenesis. The immature and markedly permeable neovascularization contributes to a reduced ability to retain the contrast medium, justifying the low iodine concentrations observed in both the arterial and venous phases. This behaviour was also observed in our study. For instance, lower iodine concentrations were found in those LI-RADS 3 nodules that subsequently progressed to LI-RADS 4 or LI-RADS 5, compared to LI-RADS 3 nodules that remained in the same category in subsequent CT follow-up scans. This may indicate a transient phase in the complex multistep process of hepatocarcinogenesis and associated neoangiogenesis, characterized by temporarily reduced functional vascularization (a parameter often linked to more aggressive or less differentiated tumors, accompanied by marked extracellular matrix deposition), which could explain the temporary reduction in iodine concentration observed in those LI-RADS 3 nodules.